



# Cross-canyon variability in zooplankton backscattering strength in a river-influenced upwelling area

Macarena Díaz-Astudillo[1,2], Manuel Castillo[3], Pedro A. Figueroa[4], Leonardo R. Castro[2], Ramiro Riquelme-Bugueño[5], Iván Pérez-Santos[2,6,7], Oscar Pizarro[8,9,10], and Gonzalo S. Saldías[1,2]

[1]Departamento de Física, Facultad de Ciencias, Universidad del Bío-Bío, Concepción, Chile
[2]Centro de Investigación Oceanográfica en el Pacífico Sur-Oriental (COPAS COASTAL), Universidad de Concepción, Concepción, Chile
[3]Centro de Observación y Análisis del Océano Costero (COSTAR-UV), Universidad de Valparaíso, Valparaíso, Chile
[4]Department of Earth & Environment, Boston University, Boston, MA, USA
[5]Departamento de Zoología, Facultad de Ciencias Naturales y Oceanográficas, Universidad de Concepción, Concepción, Chile
[6]Centro i-mar, Universidad de los Lagos, Puerto Montt, Chile
[7]Centro de Investigaciones en Ecosistemas de la Patagonia (CIEP), Coyhaique, Chile
[8]Departamento de Geofísica, Universidad de Concepción, Concepción, Chile
[9]Instituto Milenio de Oceanografía, Concepción, Chile
[10]Centro de Instrumentación Oceanográfica (CIO), Universidad de Concepción, Concepción, Chile

**Correspondence:** Macarena Díaz-Astudillo (macarenapaz.da@gmail.com)

**Abstract.** Zooplankton are a key component of food webs in upwelling systems. Their distribution is affected not only by mesoscale and climate dynamics, but also by topography and local currents. Submarine canyons that cut the continental shelf can act as conduits that transport deep nutrient-rich waters to shallower areas, promoting coastal biological productivity. Consequently, canyons facilitate the advection and accumulation of zooplankton. We aimed to describe the spatio-temporal variability in zooplankton distribution (from net samples and acoustic data) and their association with local currents, in a long and narrow submarine canyon located in the highly productive continental shelf of central Chile. The backscattering strength (Sv), a proxy for zooplankton biomass, was highly variable at a diurnal and spatial scale. Higher Sv and abundances were found during nighttime, following the classic diel vertical migration pattern. Zooplankton was not uniformly distributed within the canyon. In the surface and mid-depth layers, the canyon walls accumulated more zooplankton than the center of it, specially during the night. Within the canyon, the currents were asymmetrical and frequently changed direction. When the positive along-canyon current was more intense in the northern than in the southern slope, Sv was also higher to the north. This pattern was clearer in the section closer to the canyon head. We show that submarine canyons are highly dynamic environments where conditions can rapidly change and currents revert. Our findings suggest a possible mechanism for zooplankton retention based on the asymmetry of canyon currents and the changes in horizontal zooplankton distribution.





## 1 Introduction

In the Humboldt Current System (HCS) wind-driven upwelling is the main mechanism behind the high primary productivity levels that fuel abundant zooplankton communities (Brink, 1983; Escribano et al., 2012; Medellín-Mora et al., 2016). Other relevant mesoscale and regional drivers of zooplankton abundance and distribution in this ecosystem are coastal upwelling dynamics (Landaeta and Castro, 2002; Yannicelli et al., 2006a; Riquelme-Bugueño et al., 2012; Díaz-Astudillo et al., 2022),

mesoscale eddies and fronts (Morales et al., 2007, 2010; Pavez et al., 2010; Riquelme-Bugueño et al., 2015), changes in water-mass distribution (Aronés et al., 2009), and remote low-frequency oscillations (Díaz-Astudillo et al., 2024). Although the influence of these processes on zooplankton dynamics is relatively well known, sub-mesoscale mechanisms structuring zooplankton distribution, such as the effect of coastal currents interacting with topography (Prairie et al., 2012), have been less studied.

Abrupt changes in topography (e.g. seamounts, headlands, valleys, and submarine canyons) can lead to the retention or transport of zooplankton and micronekton through different physical mechanisms related to flows over the topography (Genin, 2004). Submarine canyons are V-shaped topographic features that interrupt the continuity of the continental shelf and/or slope for a few to hundreds of kilometers, and can be found on the continental margins of all continents (Harris and Whiteway, 2011). They modify the geostrophic flow that normally follows the direction of the bathymetric contours. The flow over submarine

canyons is dominated by advective and ageostrophic forces, leading to the generation of baroclinic tides, internal waves, and horizontal and vertical flows that enable the exchange of surface and deep waters through mixing and advection processes (Allen and Durrieu De Madron, 2009). Generally, the presence of submarine canyons in an eastern boundary continental margin significantly increases the transport of subsurface water from the slope to the inner shelf, promoting water exchange along the cross-shore axis on a relatively small spatial scale near the canyon (Allen and Durrieu De Madron, 2009; Connolly

and Hickey, 2014).

Usually, negative along-slope flows (i.e. equatorward) generate upwelling in the downstream wall of the canyon, while positive flows cause downwelling (Allen and Durrieu De Madron, 2009). In central Chile, episodes of extreme upwelling have been observed at the head of the Biobío Canyon, decoupled from wind forcing. These episodes, evidenced by the shoaling of the 10°C isotherm, occurred under weak wind conditions, with currents over the canyon in a northeast direction and negative sea

level anomalies, suggesting the passage of coastal trapped waves (Sobarzo et al., 2016). These waves intensify the topographic upwelling that occurs in submarine canyons (Sobarzo et al., 2016; Saldías et al., 2021). This and other studies prove that topographically induced upwelling and downwelling respond to both wind forcing and sea level anomalies (Wang et al., 2022).

Through the upwelling of dense deep water into shallower depths, some canyons can provide a similar nutrient input than local wind-driven upwelling (Connolly and Hickey, 2014). If the deep, nutrient-rich water reaches the photic layer for a period

long enough to promote primary productivity, canyons can become local hotspots of biological productivity and pelagic and benthic diversity (Genin, 2004; Fernandez-Arcaya et al., 2017; Santora et al., 2018). Additionally to enhanced upwelling, increased vorticity generates surface recirculation dynamics and asymmetric currents that can lead to the formation of cyclonic eddies near the canyon rim (Connolly and Hickey, 2014), which can concentrate particles or organisms. This mechanism of





aggregation has been observed in a relatively shallow submarine canyon in the Western Antarctic Peninsula, which concentrates
krill near the head of the canyon (Hudson et al., 2022).

Particle transport through canyon-mediated currents can result in the concentration of potential prey in shallower waters, thus supporting trophic interactions and high predator aggregations (Genin, 2004). Zooplankton is an essential component of the food webs of upwelling systems (González et al., 2004; Miller et al., 2010; Thompson et al., 2012; Ekau et al., 2018). Most zooplankton groups perform diel vertical migrations (DVM) within a 24 h period to avoid predation, swimming to
deeper waters during daylight hours and returning to the surface at dusk (Bandara et al., 2021). Their vertical movements can interact with coastal wind-driven or tidal currents to match or avoid offshore advection (Castro et al., 1993; Miller and Shanks, 2004; Yannicelli et al., 2006b; Shanks et al., 2014; Meerhoff et al., 2015). Because of the intensified cross-shore currents within submarine canyons, they are thought to be areas where accumulation of zooplankton is high and their advection is low (Vindeirinho, 1998). Thus, submarine canyons usually serve as foraging sites for several pelagic predators, such as whales
(Schoenherr, 1991; Croll et al., 2005; Moors-Murphy, 2014; Salgado Kent et al., 2021; Amano et al., 2023; Buchan et al., 2023), penguins (Clarke et al., 2006; Santora and Reiss, 2011; Schofield et al., 2013; Hudson et al., 2022), and fish (De Leo et al., 2012; Saunders et al., 2021), among others. Submarine canyons are also feeding, nursery and refuge sites for vulnerable ecosystems, as well as essential habitats for fish and invertebrate resources (Yoklavich et al., 2000; Sink et al., 2006; Cartes et al., 2010; Sigler et al., 2015; Santora et al., 2018). Therefore, interest in studying and protecting these coastal habitats has
grown.

The exploration and study of submarine canyons in the eastern South Pacific continental margin has mainly focused on their geological and physical dynamics, with only few studies indirectly evaluating their role in biological processes (Silva and Araújo, 2021). The Biobío Canyon (BBC) is a long canyon located in the upwelling-influenced continental shelf off central Chile. The area surrounding the BBC provides important ecosystem services (Soto et al., 2022), and is known for having
abundant aggregations of zooplankton (Yannicelli et al., 2006b; Landaeta et al., 2008) and zooplankton predators (e.g. whales, Cisterna-Concha et al. (2023)). However, the mechanisms driving its enhanced biological productivity remain largely unknown. Hence, the goals of this study are (1) to describe the intra-diurnal changes in zooplankton abundance and backscattering strength over the BBC after an upwelling event, and (2) to explore the canyon-driven physical processes underlaying zooplankton dynamics.

**2  Data and Methods**

The BBC is a river-influenced, shelf-incising submarine canyon located in the continental shelf off central Chile, to the north of the Gulf of Arauco (Fig. 1). The canyon is born near the mouth of the Biobío river, and it then zigzags for ∼40 km in the W-E direction until it reaches the continental break. From then on, it takes a SE-NW orientation and extends into the submarine trench and the abyssal plain, adding up to a total length of 134 km (Rodrigo, 2010). Its width fluctuates between 3 and 9 km,
and its depth between 20 and 1200 m.



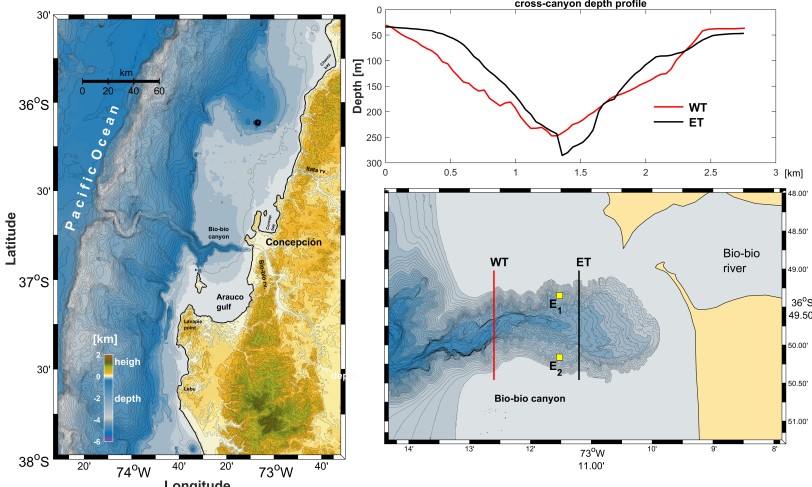

**Figure 1.** Location of the Biobío Canyon in the continental shelf and slope of central Chile. The western transect (WT) is portrayed in red, and the eastern transect (ET) in black. The bathymetric profile of both transects is shown in the upper panel. Zooplankton samples were taken at stations E1 and E2.

From July 27 to July 28, 2023, we conducted a 26 h experiment over the BBC during which we collected data on horizontal currents, acoustic backscatter, and hydrographic structure onboard the L/C Kay Kay II. Two transects were sampled 8 times each over the course of 26 h. The ship was constantly navigating from the eastern transect (ET in Fig. 1) to the western transect (WT in Fig. 1), which was located 2.1 km to the west of the ET.

On a seasonal scale, surface temperature in this area is mostly controlled by the seasonal heat flux cycle, while surface salinity is controlled by the annual cycle of the Biobío river, which outflows directly to the head of the BBC (Sobarzo et al., 2007). The Biobío river has the highest average annual discharge of all rivers in central Chile. Its annual cycle is mainly controlled by precipitation, resulting in higher discharges during the winter months (Masotti et al., 2018). Consequently, the river plume and riverine nutrient export is higher during the rainy season (June to September, austral winter) (Masotti et al., 2018). In the region, winds are mainly driven by the Pacific anticyclone with a seasonality marked by southwestern (SW) winds during austral spring-summer and northeastern (NE) winds during autumn-winter (Sobarzo et al., 2007; Ancapichún and Garcés-Vargas, 2015).

## 2.1 Hydrographic conditions

Temperature, salinity and chlorophyll-a (chl-a) profiles were obtained with a rapid-response towel CTD (Teledyne Valeport Rapidpro CTD) equipped with a fluorescence sensor. The CTD was deployed at each navigation along the transects WT and ET (Fig. 1) with a total of 14 transects for the entire sampling period. Each downward profile was obtained by releasing the instrument at free fall constant speed, and the subsequent upward profile was retrieved by the help of a winch system. Measurements comprise from the surface down to 100 m depth in the canyon area, or up to near the bottom in the area outside





the canyon, with a temporal resolution of 4 Hz for all sensors. The transects were surveyed with a maximum of 2 hours between
each one and then interpolated using the Barnes objective analysis scheme (Barnes, 1994).

## 2.2   Winds, tides and river discharge

Hourly reanalysis wind data were downloaded from ERA5 (https://cds.climate.copernicus.eu/) which is the fifth generation
ECMWF for the global climate and weather. The ERA5 data are gridded in a 0.25°x0.25° (c.a. 31 x 31 km) spatial resolution.
The nearest pixel of the grid to the location was selected to download the u (east-west) and v (north-south) components (10 m
above the surface) of the wind product (more details (Hersbach et al., 2020)). The tide data from Concepcion Bay were down-
loaded from the Sea Level Monitoring Facility (https://www.ioc-sealevelmonitoring.org/list.php), which were recorded every
1 minute. This tide gauge is part of the Chilean monitoring network from Servicio Hidrografico y Oceanografico (SHOA) of
the Chilean Navy (https://www.shoa.cl/). The discharge data of the Biobío river were downloaded from Direccion General de
Aguas (https://snia.mop.gob.cl/dgasat/pages/dgasat_main/dgasat_main.htm) which maintains a network of discharge monitor-
ing stations in Chile. Daily discharge values measured close to the Biobío river mouth were used to characterize the freshwater
input of the region.

## 2.3   Currents

Underway currents and acoustic backscatter profiles were measured during the navigation track using a downward looking 153
kHz Quartermaster Teledyne RDI Acoustic Doppler Current Profiler (ADCP). The ADCP was mounted on a stainless steel
arm on the side of the ship and fixed to have the transducers 1 m below the surface. Real-time measurements were recorded
in bottom-track mode with a maximum ship speed of 2 knots (2.5 m$^3$ s$^{-1}$). The ADCP configuration considered to acquire
currents profiles using 120 cells of 3 m bin size with 1 s pings and recorded every 4 s. The first bin was located at 4.31 m from
the ADCP and the maximum range was 350 m. Currents were recorded as earth components u (east-west) and v(north-south).
This coordinate reference was maintained along the entire analysis.
The quality control of the current profiles considered the elimination of obviously erroneous data from each circuit, a
standard criteria of goodness over 50%, flows $< 10$ m s$^{-1}$, and error less than 0.008 m s$^{-1}$. Additionally, we only considered
the profiles whose difference between the ADCP bottom track recorded speed and the GPS speed were less than 0.30 m s$^{-1}$
(e.g. Lwiza et al. (1991); Cáceres et al. (2006); Castillo et al. (2012)). The second criteria considered the correction of the
direction, since the ADCP magnetic compass is affected by the magnetic fields generated by the displacement of the vessel and
by the local magnetic deviation (Joyce, 1989; Pollard and Read, 1989; Trump and Marmorino, 1997). The residual currents
were estimated for each component (u and v) for K1 (23.93 h) and M2 (12.42 h) by the least-square fitting analysis following
the procedure suggested by Lwiza et al. (1991). The results of the harmonic analysis and the residual circulation revealed that
the tidal currents were ($< 0.05$ m s$^{-1}$).




## 2.4 Zooplankton backscattering strength

To study high resolution changes in zooplankton abundance over the BBC, we converted the ADCP's echo counts to mean volume backscattering strength (MVBS). The MVBS (hereafter called Sv, dB re 1 m$^{-1}$) is often used to observe zooplankton distribution and behavior as it gives high resolution data collected passively and simultaneously with current data (Fielding et al., 2004; Dwinovantyo et al., 2019; Cisewski et al., 2021). Sv was computed for each depth cell following the sonar equation proposed by Deines (1999) and modified by Mullison (2017):

$$Sv = C + 10log_{10}[(Tx + 273.16)R^2] - 10log_{10}L - P_{DBW} + 2\alpha R + 10log_{10}(10^{K_c(E-E_r)/10}) - 1 \tag{1}$$

In this equation *C* is a sonar-configuration scaling factor (-161.01 dB) that is specific to the RDI Workhorse QuarterMaster ADCP, *Tx* is the temperature at the transducer (°C), *L* is the transmit-pulse length (2.85 m), *PDBW* is the 10log10 of the output power (15.72 W), $\alpha$ is the depth-variable sound absorption coefficient (dB m$^{-1}$), *E* is the recorded automatic gain control (AGC or "echo counts"), and *Er* is the echo reference, determined as the minimum AGC recorded for each beam in the absence

of any acoustic signal (40 counts).*Kc* is a beam-specific sensitivity coefficient used to convert the raw echo data into dB, and it was calculated following Bozzano et al. (2013) as:

$$Kc = \frac{127.3}{Tx + 273.16} \tag{2}$$

Finally, *R* is the slant range to the sample bin (m), which uses the depth as a correction (Lee et al., 2004). Therefore, *R* is expressed as:

$$R = \frac{B + \frac{L+d}{2} + ((n-1)d) + \frac{d}{4}}{(cos\theta)}\frac{C^-}{C_I} \tag{3}$$

where *B* is the blanking distance (3.23 m), *d* is the depth cell size (3 m), *n* is the depth cell number of the particular scattering layer being measured, $\theta$ is the beam angle (20°), $C^-$ is the average sound speed from the transducer to the depth cell (1453 m s$^{-1}$) and $C_I$ is the nominal sound speed used by the instrument (1454 m s$^{-1}$). According to the the ADCP's manufacturers manual, the maximum range of acceptable data (Rmax) is defined by:

$$R_{max} = H\cos\theta \tag{4}$$

where *H* is the distance between the instrument and the bottom. All data below Rmax was eliminated, which contributed to reducing any potential seafloor noise. A minimum correlation of 25% among beams was used. If the Sv of a given beam exceeded the mean Sv of the other 3 beams by 5 dB, the bin was discarded. This process assures echo count consistency among the beams and eliminates large scatterers (Jiang et al., 2007). The 4 beams were averaged and a median filter was applied to

smooth the signal. The final Sv was then gridded at a horizontal spatial resolution of 40 m.





After carefully analyzing the vertical distribution of the scatterers, 3 layers were identified based on their average Sv and diurnal behavior: a surface layer from the surface to 25 m deep, a mid-depth layer from 25 to 100 m deep, and a deep layer from 100 m to the bottom. These layers were later used to compare the Sv and horizontal flows between transects and slopes.

## 2.5 Zooplankton sampling

Stratified zooplankton samples were taken at 2 stations (E1 and E2 in Fig. 1) during both daytime and nighttime. To avoid interfering with the continuous hydrographic and acoustic sampling that took place between 8 PM on July 27 and 10 PM on July 28 (local time), the zooplankton sampling was conducted before (2 PM on July 27) and after (1 AM on July 29) the experiment. A Tucker Trawl net (1 m$^2$ of mouth area and 300 $\mu$m of mesh size) equipped with a General Oceanics flowmeter was deployed to 100 m depth and then obliquely towed to the surface to obtain 2 stratified samples from 100-50 m and 50-0 165 m. The sample was fixed with 5% buffered formaldehyde for taxonomic analyses. In the laboratory, zooplankton groups were identified and quantified, and the abundance was standardized to individuals per 1000 m$^3$.

## 3 Results

### 3.1 Oceanographic conditions

To put the observations in the context of the environmental conditions for the region, the time series of the wind vector, 170 Biobío river discharge and Concepcion Bay tides were obtained for July-August 2023 (Fig. 2). During July-August, the winds were mainly from the south with intensities below 12 m s$^{-1}$, and the northern winds were weak (c.a. 5 m s$^{-1}$). An event of strong upwelling-favorable winds took place right before the sampling. During the ADCP measurements, winds were weak and mainly from the north. The tidal cycle was in late ebb with a range of 1 m and a river discharge of ∼1600 m$^3$ s$^{-1}$.

The water column was highly stratified due to the Biobío river input. The first 20 m of the water column had lower salinity, 175 lower temperature and higher chl-a concentration than the rest of the water column (Fig. 3 and Supplementary Fig. 1 to 4). In the upper 20 m, the ET had relatively higher chl-a and lower temperature and salinity than the WT due to its closer position to the Biobío river (Fig. 3 A and B). Below 20 m deep, the temperature and salinity diagrams show the presence of the Equatorial Subsurface Water, which is typical for the area and intrudes in the coast during upwelling conditions (Sobarzo et al., 2007). Below 20 m the WT and ET were similar in their hydrographic structure, with the greatest change around the 1025.6 kg m$^3$. 180 The WT had a slightly higher proportion of data within the 33.4 to 34 salinity range, which presented the highest chl-a values within that depth (Fig. 3 C and D).

The pycnocline was defined as the 1025.6 kg m$^3$ density contour. Its depth fluctuated through time and along the cross-canyon axis (Fig. 4). In general, the pycnocline was shallower in the ET, shifting between 15 and 35 m depth. In the WT, its depth varied between 12 and 30 m. In the WT the pycnocline was ∼10 m deeper on the southern slope of the canyon, while 185 in the ET the pycnocline had a similar depth on both walls, although it was slightly shallower on the northern slope during the



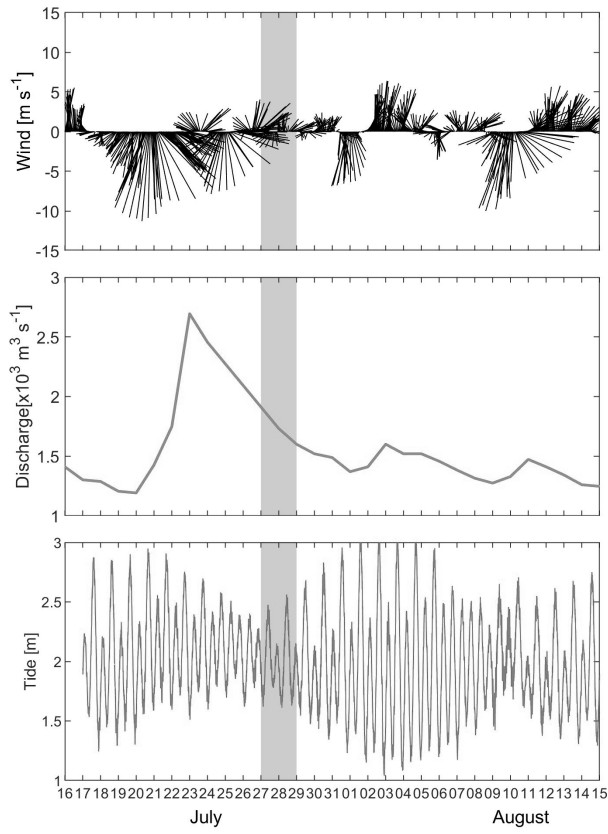

**Figure 2.** Time series of wind vectors, Biobío river discharge, and tidal cycles during the study period (in gray).

first 18 h of the study. In both transects the pycnocline got deeper through time and reached its maximum depth by the end of the study.

## 3.2 Horizontal flows over the canyon

The mean flow in both the WT and ET is shown in Fig. 5. In both transects, a mean offshore (i.e. negative) flow was observed
190 from the surface to 50 m deep in the u component. From 50 m to ∼200 m the mean flow in the WT was positive and centered in the canyon (Fig. 5 A), and positive and tilted to the northern slope in the ET (Fig. 5 B). From 200 m to the bottom the flow became negative again, although the standard deviation of the flow was high at both transects (Fig. 5 C and D). The v component showed a positive flow in the surface layer it both transects, although velocities were slightly higher in the WT (Fig. 5 E and F). The northward direction of the surface flow suggests that it possibly is a wind-driven flow. From 50 m to the
195 bottom, the flow was negative and had a high standard deviation (Fig. 4 G and H). The rise in the standard deviation (> 0.1 m





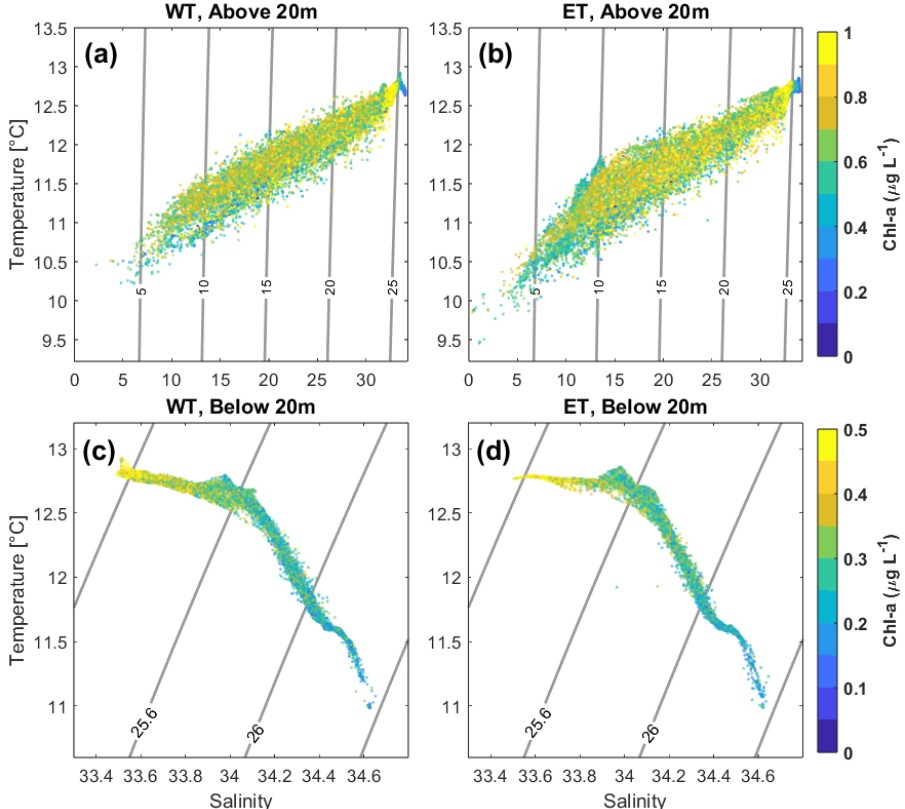

**Figure 3.** Temperature and salinity plots showing all the western (A and C) and eastern (B and D) transects together. The colors of the dots represent chlorophyll-a concentrations. The upper 20 m of the water column (panels A and B) are plotted separately than the rest of the water column (panels C and D).

s$^{-1}$) in both the cross- and along-shore currents in the deeper area of the canyon suggests that the flow is highly variable near the canyon floor.

Spearman rank correlations between the flow in the NS and SS revealed differences in the behavior of the current between the ET and the WT, and between the cross- and along-shore flow. In the ET, none of the layers had correlated u-velocities between the northern and southern slopes. Only the deep layer of the WT had a positive correlation between the cross-shore flow in the NS and the SS (Fig. 6 a, b and c). On the other hand, the along-shore velocities in the NS and SS were highly correlated in the 3 analyzed layers of the ET. In the surface layer, this suggests that the northward flow of the river plume equally affected the northern and southern sections of the canyon. In the WT, only the mid-depth layer had correlated v-velocities in the northern and southern slopes (Fig. 6). The differences in the response of the flow between the NS and the SS suggest that the cross-shore flow is much more prone to vary inside the canyon that the along-shore flow. Consequently, we further looked into the variability of the cross-shore (i.e. along-canyon) flow.




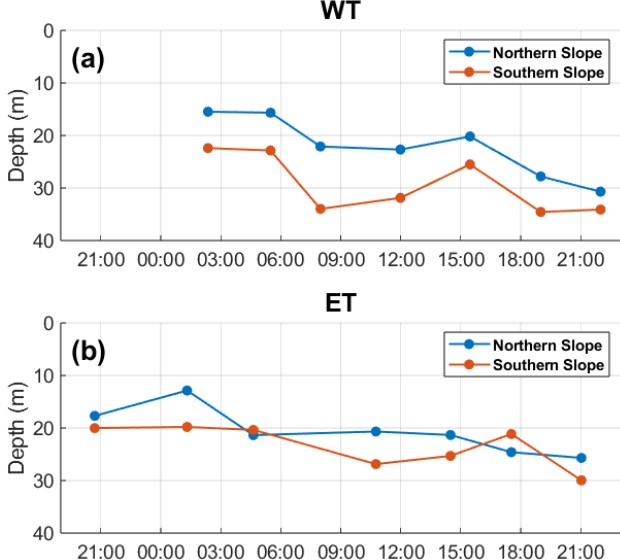

**Figure 4.** Evolution of the depth of the pycnocline through time in the WT (upper panel) and ET (lower panel). The blue line represent the NS, and the red line, the SS.

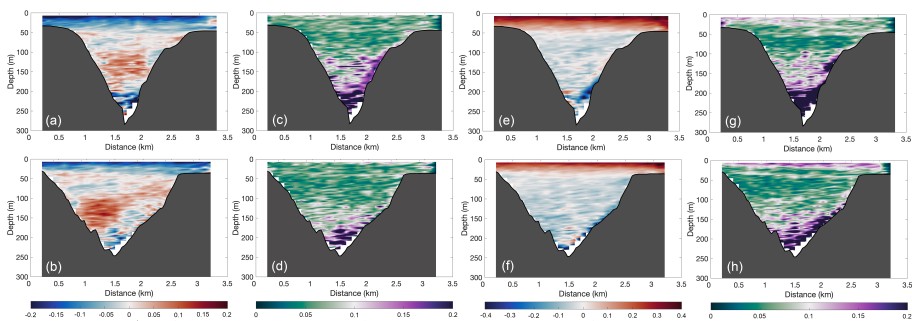

**Figure 5.** Mean cross-shore velocities in the WT (A) and ET (B) and their respective standard deviations (C and D), and mean along-shore velocities in the WT (E) and ET (F) and their respective standard deviations (G and H).

We identified the section of the canyon where bottom depth was >60 and <120 m deep. We then averaged the cross-shore velocity to see its evolution through time and by depth, at each slope. In both the NS and SS of the WT, the first 40 m of the water column had negative velocities throughout the study period. From 40 to 120 m the velocities were predominantly positive

210  in the NS, except during the first 3 hours and at the end of the experiment, when the velocities shifted and became strongly negative in the NS (Fig. 7 A). In the SS, velocities in the 40-120 m deep section remained variable, alternating between periods of positive and negative velocities (Fig. 7 B). Consequently, the difference in the velocity magnitude between both slopes also





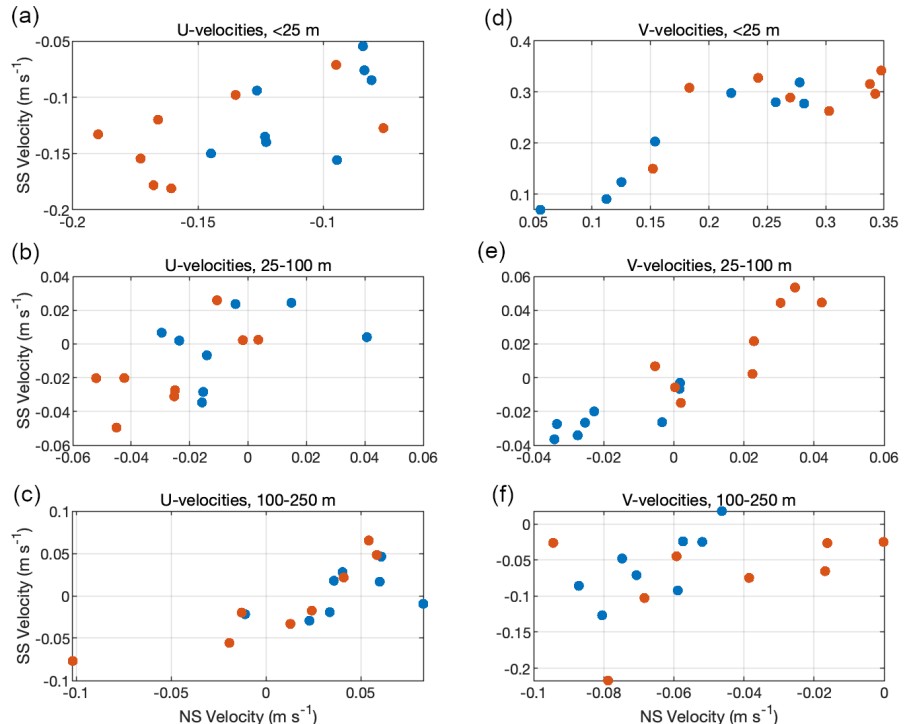

**Figure 6.** Correlations between the mean flow in the northern slope (NS) and southern slope (SS) for both the WT (red dots) and the ET (blue dots), by layer. When significant, the correlation coefficient and p-values are shown.

alternated between periods of positive and negative differences, indicating a stronger current in the southern slope at the end of the experiment (Fig. 7 C). The ET also had mostly negative velocities in the upper 40 m of the water column, both in the NS and SS. In the NS, the rest of the water column had mostly positive velocities throughout the experiment (Fig. 7 D), while the SS presented mainly negative velocities (Fig. 7 E). The differences between both slopes was variable in the upper 40 m of the water column, and mostly positive in the deeper section, indicating stronger along-canyon currents in the northern slope (Fig. 7 F).

### 3.3 Zooplankton spatial and temporal variability

The Sv over the canyon had high temporal, vertical and spatial (between transects and slopes) variability. Some of the changes observed in the Sv were also observed in the zooplankton samples. The more evident pattern present in the Sv sections was the higher observed Sv in the night transects, both at the WT and ET (Fig. 8 and Supplementary Fig. 5). The biological samples taken near ET showed that most of the zooplankton groups also exhibited higher abundances during the night (Fig. 8). The cross-canyon Sv sections also showed differences in Sv between the northern and southern slopes, which changed with time. Zooplankton samples also showed this pattern. By the end of the experiment, higher abundances were found on the southern



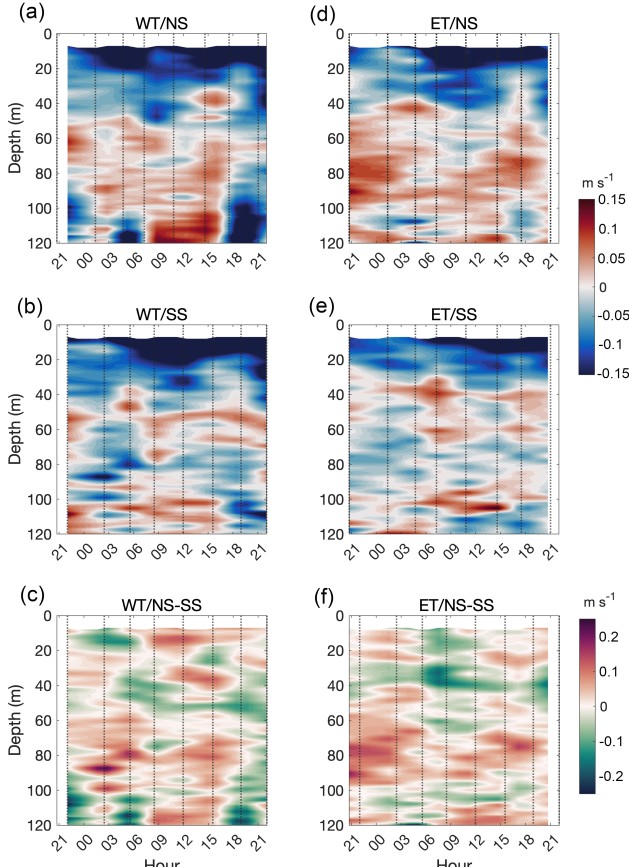

**Figure 7.** Mean cross-shore currents by depth and time, in a section of the canyon walls where bottom depth was 60-120 m deep. The currents over the northern slope (NS, upper panels) and southern slope (SS, middle panels) of both the WT (left panels, A and B) and ET (right panels, D and E) are shown. The difference in the current magnitude between the NS and SS is shown in panels C (WT) and F (ET). The dotted lines overlaying the plots represent the time of each transect.

slope, and Sv was also significantly higher in the southern wall (Fig. 8). These results validate the use of the acoustic data as a proxy for zooplankton abundance.

The canyon slope and diurnal differences in backscattering strength changed not only over time, but also with depth. The surface layer had high intra-diurnal temporal fluctuations in the Sv values, with evident higher Sv during the night. In the WT, the difference in Sv between the lowest daytime and the highest values was $\sim$12 db re 1 ms$^{-1}$. In the ET, this difference reached $\sim$18 dB re 1 ms$^{-1}$ (Fig. 9 A and B). In the ET, the mid-depth layer had the highest mean Sv values, while in the WT its values were similar than those of the surface layer. In both transects, the U-shape of the lines representing the mean Sv evidenced an effect of the canyon shape on zooplankton abundances (Fig. 9 C and D). This translated in higher abundances over the northern and southern slopes than in the central area of the canyon. In the mid-depth layer, the diurnal pattern in Sv





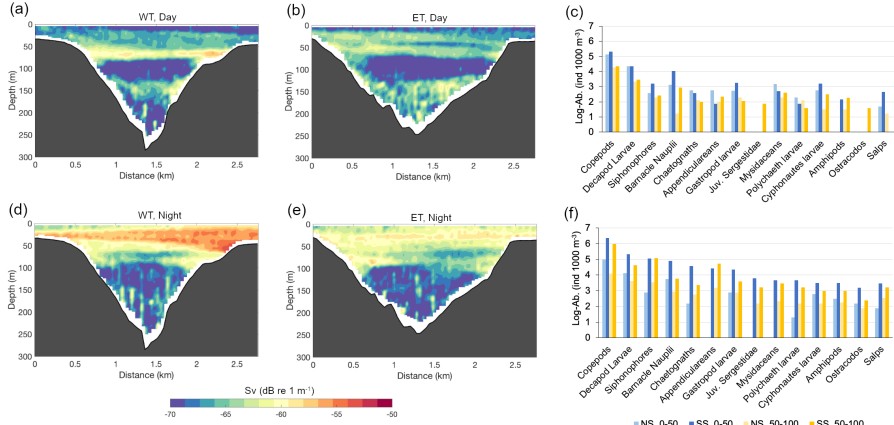

**Figure 8.** Selected Sv sections over the western transect (WT) and eastern transect (ET). A: first diurnal WT. B: first diurnal ET. D: last night WT. E: last night ET. Standardized zooplankton abundances by group (collected in stations E1 and E2 close to the ET) are shown in panel C for samples collected the day prior to the beginning of the experiment (daytime), and in panel F for samples collected after the end of the experiment (nighttime). NS: northern slope, SS: southern slope.

was clear (especially over the canyon slopes), but less intense than in the surface layer. The deep layer showed little change in Sv with time. The evident diurnal differences found in both the surface and mid-depth layers were not found in the deeper layer, where the diurnal differences were negligible (Fig. 9 E and F). The effect of the canyon shape was also present in the deeper layer, observed as higher Sv near the canyon walls, and lower Sv in the central area of the canyon.

To compare the Sv between the northern and southern canyon slopes, we repeated the method previously done with the
u-velocities. The section of the canyon where bottom depth was >60 and <120 m deep was identified to then average the Sv to see its evolution over time on each slope. We also calculated the Sv difference between the northern and southern sections and correlated that with the horizontal flows in each layer. Several differences in the vertical distribution of zooplankton were observed between slopes, at both transects. The NS of the WT had high Sv in the 20-80 m water column section during the entire study period (Fig. 10 A). The SS of the WT consistently showed a layer of ∼20 m width with high Sv values,
which deepened during the day and became shallower and wider during the second night (Fig. 10 B). The intensity of the backscattering strength was higher in the SS, thus the difference in Sv between the NS and SS was mostly negative throughout the study Fig. 10 C). In the NS of the ET, high Sv values (>63 dB re 1 ms$^{-1}$) were present in almost the entire water column during the first 12 h of the study (Fig. 10 D). Later in the day, Sv decreased and low values were observed within the first 50 meters of the water column. Surface Sv increased again near 8 PM. In the SS of the ET high Sv values were observed only
between the 20-50 m layer in the first 12 h of the study. Near 9 AM, that layer deepened and its intensity decreased (Fig. 10 E). The difference in Sv between the NS and the SS of the ET was ∼10 dB re 1 m$^{-1}$ between 50 and 120 m at the beginning of the study, which agreed with strong differences in the along-canyon currents (i.e. higher velocities in the northern wall of





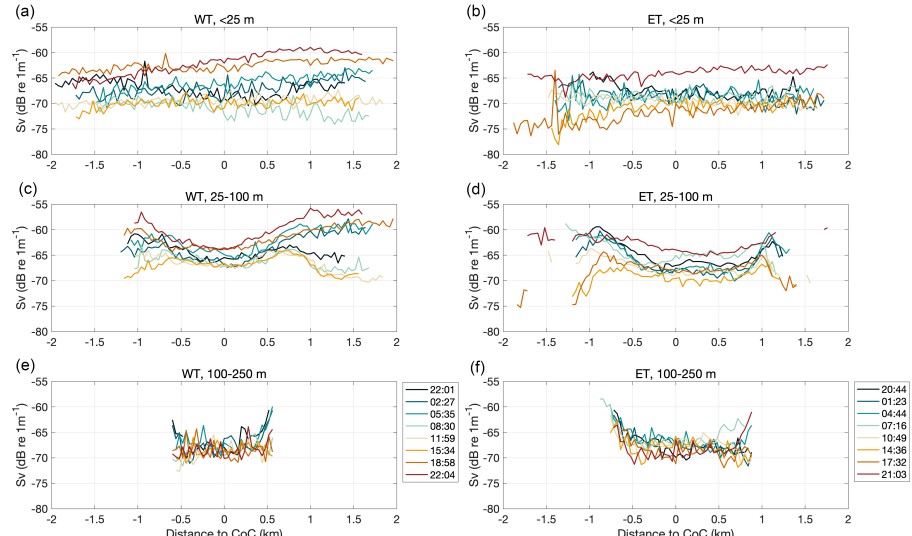

**Figure 9.** Mean Sv in the surface (upper panels), mid-depth (middle panels) and deep (bottom panels) layers along the WT and ETs. Each of the 8 tracks along the western (left panels) and eastern (right panels) transects are represented by different colors. Distance was referenced to the center of the canyon (CoC), identified as the deepest point of it, in order to facilitate the comparison between the northern and southern slopes. Negative distances represents the northern slope, and positive distances the southern slope.

the ET). Although this difference decreased with time, the NS consistently had higher Sv within that layer compared to the SS (Fig. 10 F).

## 4   Discussion

The "canyon hypothesis" suggests three main mechanisms by which submarine canyons promote local biological productivity (Hudson et al., 2022; Genin, 2004). The first is related to the fertilization of surface and subsurface layers through the advection of deeper nutrient-rich waters to the surface (i.e. "topographic upwelling), which should last long enough to allow phytoplankton and zooplankton populations to reproduce and increase in abundance. The second mechanism involves the generation of a subsurface eddy, which causes isopycnal doming and consequently leads to upward water transport and/or enhanced particle retention. The third mechanism involves the physical retention and aggregation of organisms due to the interaction of currents with topography and DVM. Worldwide, there is little direct evidence of these mechanisms because of the challenges of studying the highly dynamic processes that take place in submarine canyons. Nonetheless, we present evidence to prove that the BBC plays an important role in zooplankton distribution variability on a diurnal scale by means of the third and a possible fourth mechanism.





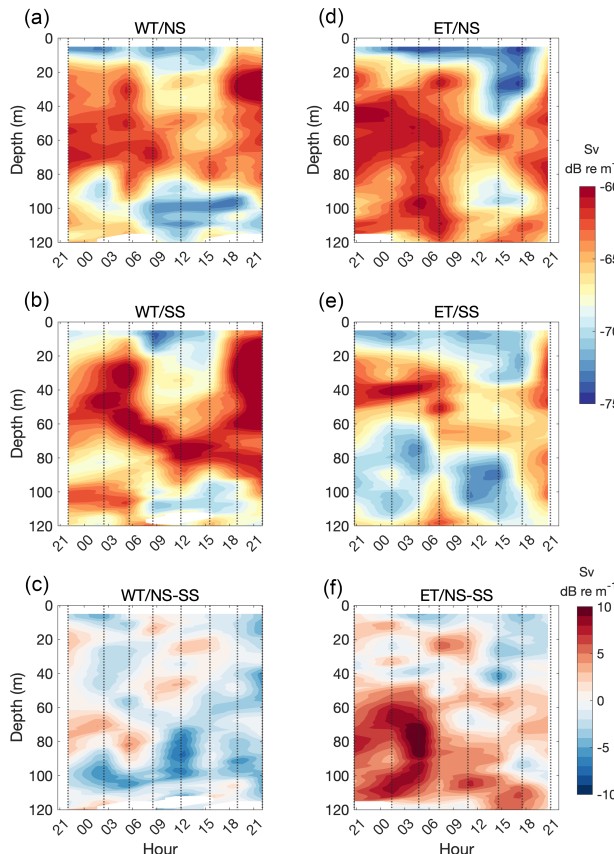

**Figure 10.** Mean Sv and Sv difference in a section of the canyon walls where bottom depth was 60-120 m deep. The northern (upper) and southern (middle panels) slopes of both the WT (left panels, A and B) and ET (right panels, D and E) are shown. The difference in Sv between the NS and SS is shown by depth and time in panels C (WT) and F (ET). The dotted lines overlaying the plots represent the time of each transect.

## 4.1 The interaction of diel vertical migrations with abrupt topographies

The intra-diurnal changes in Sv within the canyon were mostly linked with zooplankton diel vertical migrations (DVM), while the spatial changes appeared to be associated with the canyon presence and shape. The accumulation of organisms near the canyon slopes was evident in the U-shape of the Sv lines, as higher Sv was found in both the northern and southern walls compared to the central region of the canyon. This pattern might hint at an interaction between zooplankton migration the steep topography. In the general DVM pattern, zooplankton migrates to deeper layers at dawn and swim up to the surface at dusk (Forward, 1988; Hays, 2003). Shallow topographies might block the descend of zooplankton towards deeper surrounding waters, causing an accumulation of organisms over shallow areas, a mechanism called "topographic blocking" often evidenced over seamounts (Aarflot et al., 2019; Mohn et al., 2021). Another type of topographic blocking is found when horizontal






currents transporting organisms encounter a steeply sloping bathymetry, such as shallow banks (Isaacs and Schwartzlose, 1965) or the continental shelf break (Mackas et al., 1997). Submarine canyons have long been proposed as features prone to generate topographic blocking (Genin, 2004), as bathymetry changes abruptly in the canyon walls.

Additionally, zooplankton DVM can interact with alternating vertical currents to promote zooplankton retention. We often found opposite directions in the flow over depth (see Fig. 5), as well as differences in Sv between vertical layers (see Fig. 10

and Supplementary Fig. 5). In coastal environments, alternating flows are usually associated with tidal currents or estuarine circulation (Valle-Levinson et al., 2014; Meerhoff et al., 2015). In this study, we found alternating flows that had little association with the tidal cycle, but were possibly explained and enhanced by the presence of the canyon. The zooplankton community was composed of several groups that are strong migrants, such as decapod larvae, amphipods and copepods (Mackas et al., 2005; dos Santos et al., 2008; Escribano et al., 2009; Bandara et al., 2021). These organisms may exploit the fluctuating vertical

flows to avoid offshore advection by migrating between layers with opposed directions, a well-documented mechanisms of zooplankton retention in tidal and estuarine systems (Castro et al., 1993; Hill, 1998; Poulin et al., 2002; Emsley et al., 2005; Kimmerer et al., 2014).

### 4.2 Canyon-mediated advection and retention

We also found striking differences in zooplankton abundance between the NS and SS, in both the acoustic and in situ sampling, which were highly variable and alternating over time. Apparently, these differences were associated with the contrasting cross-

shore flows on either side of the canyon. Observational and numerical modeling studies have shown that submarine canyons impact and modify coastal circulation (Sobarzo et al., 2016; Saldías and Allen, 2020). The presence of a shelf-incising canyon in a western continental margin generates a flow "dipole" where an along-canyon inshore flow is found in the downstream canyon wall, and an outflow in the upstream wall (Allen and Durrieu De Madron, 2009; Vergara et al., 2024). A recent study

modeled the influence of the BBC on the coastal circulation of the Arauco Gulf during upwelling and downwelling events, using high-resolution hydrodynamic simulations (Vergara et al., 2024). They evidenced the formation of the dipole in the mean cross-shore flow field. The downstream side of the canyon (i.e. southern slope) had positive cross-shore velocities, meaning and inflow through the canyon, while the upstream side showed an offshore flow. Although this pattern was present during both upwelling and downwelling events, the inshore flow was stronger when winds were favorable for upwelling. They also

evidenced the advection of dense deep water over the shelf under upwelling conditions, which is an indicator of topographic upwelling.

The average cross-shore current showed an inshore flow of water through the canyon, which was clearly tilted to the northern slope in the ET, following the theoretical dipole pattern. This circulation pattern makes sense considering that the study took place after a short but intense event of upwelling-favorable winds. During most of the study period, the cross-shore flow in the

40-120 m layer was positive in the northern slope of the ET, and negative or alternating in the southern slope. This difference in the magnitude and direction of the currents was more evident during the first 6 h of the study, matching the shallowest depths of the pycnocline. The highest Sv difference was also found during that period of time, when Sv was ~10 dB re m$^{-1}$ higher in the NS of the ET compared to the SS. Thus, the inshore flow over the northern wall of the ET agreed with a shallower



pycnocline, higher zooplankton backscattering strength, and high zooplankton abundances. In the WT, the farthest from the

canyon head, the coherence between the cross-shore flow and the acoustic backscatter was less clear. Nonetheless, there was a strong difference in Sv between the slopes of the WT by the end of the experiment, which was also evident in the zooplankton samples, as higher abundances were found in the SS. In the same period (the last 6 h of the experiment) a strong offshore flow was observed in the NS that might explain the decrease in Sv on the NS. Overall, the differences in Sv between the northern and southern wall, along with their alternations (Fig. 7 and 10) during the 24 h cycle suggest that organisms may use the

asymmetrical currents to retard the advection and/or enhance retention within the canyon. This mechanism acts similar to the interaction of DVM with opposed vertical flows, but in the horizontal axis, and would result in a neutral net flux of zooplankton.

While our findings suggest that canyon currents promote an asymmetrical advection of zooplankton, more efforts are needed to confirm these findings. The variability in the flows and in Sv was high and the topography adds complexity. In the ET, the along-shore flows over the NS and the SS were highly correlated due to its proximity to the Biobío river mouth. In the WT

only the mid-depth layer had correlated along-shore velocities in the NS and SS. Regarding the cross-shore velocities, only the deeper layer of the WT showed a correlation between the flow in the NS and SS. The horizontal currents inside the canyon not only differed between walls, but also shifted in direction within a period of less than a day. Hence, longer time series together with simultaneous zooplankton sampling at both canyon walls could resolve the net transport of zooplankton.

### 4.3 Ecological implications

Some submarine canyons have substantial scientific evidence about their role in the formation of zooplankton aggregations. The Monterey Canyon, in the California Current System, is one of the best studied. This canyon is a known foraging cite for large cetaceans (Schoenherr, 1991), and is one of the main habitats for krill, in what has been defined as an ecologically critical canyon network (Santora et al., 2018). Submarine canyons of the HCS have only recently begun to be explored in depth. Earlier research on the Itata Canyon (a relatively large canyon located 60 km north of the BBC) found higher abundances

of several crustacean larvae close to the shore in the survey transects conducted over the submarine canyon. Outside of it, larvae were more abundant offshore (Yannicelli et al., 2006a). Although the dynamics inside the canyon were not described, this was an indirect sign of potential inshore transport facilitated by the canyon. New research has suggested that a recently discovered and relatively small canyon might explain the high concentration of whales and krill in a known and protected marine reserve in northern Chile (Buchan et al., 2023). To our knowledge, this is the first study that attempts to combine

simultaneous observations of zooplankton aggregations and measurements of canyon-induced currents to elucidate the physical mechanisms driving zooplankton dynamics.

Our findings highlight the role of the Biobío Canyon as a potential key player in shaping local zooplankton distributions through mechanisms such as asymmetric advection, topographic blocking, and enhanced particle retention. These processes, driven by complex interactions between canyon morphology, hydrography, and circulation patterns, emphasize the importance

of submarine canyons as ecological hotspots in the upwelling system of the HCS. However, given the dynamic nature of canyon processes and the temporal limitations of our study, longer-term and higher-resolution datasets are needed to fully understand these mechanisms. Future efforts aiming to elucidate the variability of flows and organisms distribution over sub-



marine canyons should consider the installation of multiple arrangements to cover the spatial variability existing in canyons. Expanding research efforts on submarine canyons, particularly in under-explored areas with high socio-ecological importance, is crucial for unveiling their ecological significance and their role in regional productivity. These insights are essential not only for the advancement of scientific knowledge but also for informing conservation and management strategies in these biologically rich and ecologically significant habitats.

## 5   Conclusions

We aimed to describe the spatio-temporal variability in zooplankton distribution and currents within a long and narrow submarine canyon. We found evidence to prove that the canyon influenced zooplankton distribution and abundance, all in a period of less than one day. The experiment took place after an event of upwelling favorable winds. The water column was highly stratified because of the Biobío river output. The horizontal flows within the canyon were highly variable and had high standard deviation. The mean currents showed an entrance of water towards the coast through the northern wall of the canyon in the ET. However, currents changed rapidly in direction at both canyon walls, which resulted in non-correlated current velocities between the two surveyed transects and between the canyon slopes. There were differences in the flow velocity and direction between vertical layers, and also between canyon slopes. Zooplankton abundance also changed through time and space. At the beginning of the study, abundances were higher in the northern slope, which reverted by the end of the study. The same pattern was observed in the Sv, which was always higher near the canyon walls than in the center of the canyon. In general, the SS of the WT had higher Sv than the NS, while the opposite was found in the ET. In the ET, the higher Sv in the NS coincided in time and depth with a difference in the current direction, which was positive in the NS and negative in the SS. Thus, the asymmetrical horizontal currents possible caused the horizontal differences in zooplankton distribution. This interaction between zooplankton and opposed and alternating canyon flows might promote their retention and aggregation. Our findings demonstrate that submarine canyons are highly dynamic habitats and highlight the need to study these key ecosystems, specially in areas that provide essential ecosystem services.

*Author contributions.*   MD-A: Conceptualization, data curation, formal analysis, investigation, methodology, visualization, resources, writing-original draft preparation. MC: Formal analysis, investigation, methodology, visualization, writing-review & editing. PF: Formal analysis, data curation, visualization, writing-review & editing. LRC: Resources, supervision, writing-review & editing. RR-B: Investigation, supervision, writing-review & editing. IP-S: Investigation, resources, supervision. OP: Resources, writing-review & editing. GSS: Conceptualization, investigation, methodology, funding acquisition, project administration, resources, supervision, writing – review & editing.

*Competing interests.*   The authors declare that they have no conflict of interest.



*Financial support.* MD-A was supported by the ANID FONDECYT grant 3230183. GSS was supported by FONDECYT grant 1220167. IPS was supported by FONDECYT grant 1211037 and CIEP R20F002. MD-A, GSS, IP-S and LRC were supported by COPAS COASTAL ANID FB210021.

*Acknowledgements.* The authors are grateful to the crew members who supported the field sampling and to the Centro de Instrumentación
Oceanográfica (CIO) for providing the ADCP used in this study.



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
