# Peer review of "Cross-canyon variability in zooplankton backscattering strength in a river-influenced upwelling area"

_EGUsphere, 2025_

## Author Comment (AC2)

**REVIEWER #2**

This is an interesting study based on a very small dataset consisting of 16 transects with a shipboard 150khz ADCP and 4 zooplankton tows over the course of 26 hours. Although the dataset is limited in spatial and temporal extent the authors' analysis provides some interesting insights into biophysical processes in a canyon within an upwelling regime.

R: We are very thankful for the positive opinion on our work, and we appreciate all the comments and suggestions that helped us improved it. Below you will find our answers to all your comments.

I think the authors are describing a mechanism for zooplankton accumulation in the canyon not just retention? This concept with in the literature context of canyon funneling of zooplankton should be included in the scope of the paper.

R: Thank you for the comment. We improved our discussion to include zooplankton aggregation, and not just focus on retention. We have incorporated a description on how DVM interacted with topography to aggregate zooplankton. Nonetheless, it is our understanding that canyon funneling refers to the process by which sediment, nutrients, and other materials are transported downslope from continental shelves and slopes, into the deep ocean, mainly by deep currents or gravity-driven flows (for example, https://doi.org/10.1016/0079-6611(65)90028-5, https://doi.org/10.1016/j.marpetgeo.2014.02.016, and references therein). This is fundamentally different from the mechanism we are evidencing (which involves asymmetric currents and DVM). Thus, we believe that including canyon funneling as a concept could confuse the readers.

It would be helpful to include a dashed line marking local sunrise and sunset in all figures that display time and all figures should be bigger

R. Thank you for the suggestion. We have now indicated the time of sunrise and sunset in Figures 3, 6, 9 and S6.

Conceptually my biggest concern is the use of the term topographic blocking , which is actually quite specific and well defined in the long history. Topographic blocking specifically refers to vertical migrators whose sunrise descents are blocked by seafloor with depths shallower than their preferred daytime depths. The authors' data does show some evidence for this at WT with what my best guess is for local sunrise (both sunrise and sunset times need to be explicitly stated in the text and marked in

figures). Eg discussion ~l272. I think canyon funneling - where currents advert and accumulate animals at the canyon head should be explicitly discussed and relevant literature included new and old (eg greene et al 1988). This second type of topographic blocking is not topographic blocking but rather physical aggregation through advection toward the seafloor. Overall time needs to be examined in more detail for the biological analysis and interpretation. Sunrise and sunset are important and will dramatically influence the density and distribution on zooplankton in the canyon. As is, the times of sunrise and sunset are not even explicitly mentioned.

R. We appreciate this comment as it points out an important topic in our paper. As the reviewer indicates, we have some straightforward evidence for topographic blocking at some specific moments during our sampling. We have described this phenomenon with detail in the Discussion. We still want to point out topographic blocking as a possible mechanism for zooplankton aggregations in this canyon, even knowing our dataset is limited. Regarding canyon funneling, we discarded this mechanism because is mainly associated with bottom currents and may cause some confusion. Our results shows that zooplankton aggregations were mostly present in mid-depth and surface layers.

Line specific comments

Figure 1 the color scheme for the bathymetry map is very confusing. There should not be two whites in one gradient. I suggest a single yellow to white gradient for land and a single gradient from white to dark blue for the seafloor. The cross canyon - rofilemis very useful. I would suggest adding the approximate locations of the zooplankton sampling stations.

R. Thank you for the suggestion. We modified the color scale of Figure 1. Please note that the location of the zooplankton sampling stations were already indicated in Figure 1.

L96 casts not transects?

R. After reviewing the sentence, we confirm that transects was correct, as a total of 14 transects were carried out.

L98 cite supply table 1

R. We cited Supplementary Table 1 as the Reviewer suggested.

Section 2.4 Small fish with swim bladders will also scatter at 153khz - these results are not only for zooplankton.

R. The Reviewer makes a good point with this observation. Fish with swim bladders do scatter at frequencies between 100-200 kHz, and part of the signal might come from them. Nonetheless, research shows that both individual target strength and volume backscatter strength of fish with swim bladders is higher than smaller fluid-like organisms, at the used frequency (150 kHz), and that fish usually scatter ar lower frequencies (eg. doi.org/10.1093/icesjms/fsp262, doi.org/10.1121/10.0000594). Our methodology, which includes filtering all Sv higher than -40 dB re 1m$^{-1}$, assures we are getting mostly signal from zooplankton and micronekton. However, we can't assure there is no signal coming from other groups, so me included the following in section 2.4: "Although this methodology does not completely filter the signal of bigger scatterers, such as small fish with swimbladders, it does reduce their contribution the total volume backscattering strength."

L156 mark these 3 layers with dashed lines in all relevant plots for easy reference

R: We included horizontal lines in Figure 7.

Section 2.5 more details needed. Why not include a metadata ramble for the zooplankton sampling, location, depth, time, length of tow, volume sampled, biomass etc. were there 4 samples or 2?

R. Thank you for your suggestion. We included details about the biological sampling in the new Supplementary Table 2.

How can you get 2 depth discrete samples? Did the Tucker trawl have 2 nets or 2 cod ends such that one was closed at a specific depth? How was sampling depth monitored? Were two separate casts done to sample the different depths? How did you avoid sampling on the way down and up? Opening and closing nets?

R. The Tucker Trawl net has the potential to sample 2 strata in 1 tow, as it has 3 nets that open and close with a trigger mechanism. The net is towed to the maximum depth (in this case, 100 m) with the first net open. That provides an integrated sample that was not used in this study. Then, a trigger mechanism is activated with a drop messenger to close the first net and open the second. Once the second net is open, we begin towing the net upward to cover the first stratum. The trigger mechanism is activated again once we hit the target depth (in this case, 50 m), so the second net closes and the third opens. We then tow the third net to the surface to collect the sample for the 2$^{nd}$ stratum. These details are not included in the Methodology because it is standard procedure when using a Tucker Trawl net.

L179 the 1025 'isopycnal' add isopycnal

R. "Isopycnal" was added as the Reviewer suggested.

Fig 5 caption would suggest defining positive cross shore direction and along shore direction in caption (onshore and equatorward)

R. The caption was improved to include the interpretation of positive/negative velocities in both axes.

Figure 6 p values and correlation values referenced but not shown in figure or written in text. These should be added

R. We apologize for this mistake. An earlier version of Figure 6 had the p- and rho-values for the correlations, but they got lost during figure editing. We uploaded an updated version of Figure 6.

Suppl fig 5 is very useful but needs times, WT/ET labels, sunrise and sunset marked or identified in some way, NS/SS identified.

R. We acknowledge this figure was lacking some labels and appreciate the suggestion. Even though Supplementary Table 1 had some of the information (such as each transects time), we included a label in the title of each subplot to identify which were WT or ET transects. Additionally, we marked those transects that took place near sunset and sunrise. We also indicated in the caption that, for all transects, the left side is the northern slope, and the right side is the southern slope.

It looks like in WT transects there is some evidence of sunrise accumulations of scatterers over the canyon walls. This would be evidence for topographic blocking as mentioned in the discussion. Because topographic blocking is specifically mentioned, these observations should be included in the text.

R. Than you again for this suggestion. We included these observations as evidence for topographic blocking in the Discussion.

L226 which results? Also a 1m tucker trawl likely would not do very well catching fishes but that doesn't mean that they aren't contributing to the sound scattering.

R. We improved the section to clarify our point. We tried to evidence that the differences in zooplankton abundances estimated from the nets showed the same pattern that the Sv. That is, that higher abundances were found in the southern slope of the canyon in both the standardized abundances and the Sv. Regarding the comment on the potential fish scattering, as we said in a previous comment, we acknowledge that the signal might have some contribution from other scatterers, including small fish, but this should be minimal considering the frequency and the methods used to avoid signal from bigger scatterers.

It would be useful to combine figures 7 and 10 in order to be able to look at currents and mean sv side by side, especially given the authors arguments that the currents are structuring the zooplankton distribution.

R. While we acknowledge the usefulness of merging Figures 7 and 10, we chose to keep them separate to maintain the logical flow of the Results section. Presenting them sequentially (first currents and volume backscattering strength) allows the reader to follow our reasoning from dynamic physical processes to their relationship with the acoustic observations. Merging them would necessarily imply to change the structure of the results.

Figure 8 add NS and SS as well as dashed lines for the 3 depth layers referenced. Consider

R: We appreciate the suggestion. We included labels to point out the northern and southern slope (NS and SS), as well as horizontal lines to indicate the vertical layers used for the later analyses, in what now is Figure 7.

Figure 9 could be very useful with just a few changes: color code the times in a logical way - make all the nighttime transects blues/greens and the daytime transects reds/yellows then you will see that there is an interesting temporal pattern in these data! There are interesting changes in how topography influences sv over time - this may also strengthen the topographic blocking argument and should be added to the results in l232

R: Thank you for your comment and suggestion. We changed the color scheme in a way that all daytime transects are red/yellow, and nighttime transects are blue/green.

L240 was the fact that sv are on a log scale taken into account before averaging

R: Thank you for pointing this out. As in other studies (see for example Fielding et al. 2004 doi:10.1016/j.icesjms.2003.10.011, Benoit-Bird and Waluk 2020 doi:10.1121/10.0000594, among others) we calculated the geometric mean of the MVBS to provide relative changes through time and depth. This procedure considers that the MVBS is a logarithmic measure of backscattering cross-section per unit volume. We have included this detail in the Methods section.

L245 I disagree with this Interpretation. SS of WT has a 20m thick layer during the day that greatly expands in thickness during the night

R: Thank you for sharing your point of view. Although we had explained that the layer got wider during the night, we improved the line to make it clearer, following the Reviewer suggestion. The line now goes: "The SS of the WT showed a layer of ~20 m width with high Sv values that deepened during the day, and became shallower and increased in width (up to ~100 m) during the night (Fig.10 B)".

L284 seems odd to have references here -is this not the data presented in this manuscript? Mackas 2005 was not in BBC for example.

R: Thank you for pointing that out. We acknowledge that the line was confusing. We improved it to make it clear that the references are there to prove that the groups we found are known to be strong migrants.

L285 don't the data presented here show this? One thing that was missing is the specific link over time between zooplankton layers and currents. What does your data tell you about how a migrating zooplankton in each of the 3 layers is adverted around the canyon over the 24 hrs? Where did a zooplankton who vertically migrated but was advected in the horizontal by the currents end up? Was there evidence for a funneling effect and a net transport into the inshore /canyon head direction? Were they advected in a circle and so were effectively retained ? How about a non migrating plankton at each depth. Making these points with given animal depths at different times and DVM behavior would make the biophysical connection explicit

R: We really appreciate this comment. We explicitly linked zooplankton Sv at certain depths and layers with currents at the same time and depth. We have now incorporated scenarios for difference cases, to provide direct evidence for topographic blocking and retention.

---

## Author Response (AR2)

Dear Editor,

Please find attached the new versión of the manuscript with a revised Figure 1.

Best regards,

Dr. Macarena Díaz-Astudill on behalf of all authors.